# The Effects of COVID-19 Virtual Learning on Body Fat and Insulin Resistance in Adolescents with Overweight or Obesity

**DOI:** 10.3390/children10081398

**Published:** 2023-08-16

**Authors:** Lindsay M. Stager, Casie H. Morgan, Caroline S. Watson, Skylar Morriss, Barbara A. Gower, Aaron D. Fobian

**Affiliations:** 1Department of Psychology, University of Alabama at Birmingham, Birmingham, AL 35294, USA; lmstager@uab.edu (L.M.S.);; 2Department of Social Work, University of Alabama at Birmingham, Birmingham, AL 35294, USA; 3Department of Nutrition, University of Alabama at Birmingham, Birmingham, AL 35294, USA; 4Department of Psychiatry and Behavioral Neurobiology, University of Alabama at Birmingham, Birmingham, AL 35294, USA

**Keywords:** COVID-19, adolescence, body fat, insulin resistance, obesity

## Abstract

**(1) Background**: COVID-19 virtual learning reduced structural supports for adolescent physical activity and diet, threatening metabolic health, especially in teens with overweight or obesity (OWOB). **(2) Methods**: Adolescents (N = 14) with OWOB completed fasting blood draws (measuring insulin resistance, IR) and Dual Energy X-Ray Absorptiometry (DXA, measuring total body fat percent, TBF%) pre-COVID-19 and during COVID-19. Changes in TBF% and IR were calculated (1) pre-COVID-19 and (2) from pre-COVID-19 to during COVID-19. Age and body mass index (BMI) percentile-matched data assessed normative changes across similar, non-COVID-19 time periods. Paired *t*-tests compared TBF% change pre- to during COVID-19 with (1) TBF% change pre-COVID19 and (2) TBF% normative change. Two ANCOVAs compared IR change pre- to during COVID-19 with (1) IR change pre-COVID-19 controlling for BMI z-score and difference in time between assessments and (2) normative change in IR controlling for sex/race. **(3) Results**: The TBF% change pre-COVID-19 and the normative change were similar. The TBF% increased more (~six percentage points) during COVID-19 compared to normative change (*p* < 0.01). During COVID-19, IR increased more (~2.5 units) than change pre-COVID-19 (*p* = 0.03) and increased more (~3.5 units) than normative change (*p* = 0.01). **(4) Conclusions**: TBF% and IR increased exponentially during COVID-19 in teens with OWOB compared to pre-COVID-19 and normative changes.

## 1. Introduction

The coronavirus disease 2019 (COVID-19) pandemic and subsequent call for social distancing measures required that adolescents across the United States transition from in-person classes to virtual learning for the end of the 2019 school year and, for some individuals, the beginning of the fall 2020 school year [1]. While these measures were necessary to protect immediate health, they may have had adverse long-term effects on adolescents’ wellbeing with regard to weight gain and metabolic health [2]. Notably, these changes extended adolescents’ summer schedules and limited the structural supports provided during the school year for physical activity and diet, such as structured daily routines related to in-person classes and school activities and the provision of nutritional breakfasts and lunches. Reductions in such supports are common over summer vacation and can cause adolescents to gain more weight during the summer as compared with the school year. Adolescents with overweight or obesity (OWOB) are at particularly elevated risk for this phenomenon, as they may be less likely to engage in structured recreational activities and consume balanced meals during the summer [3]. Given the established risk for weight gain associated with not being in school during the summer, the extension of adolescents’ summer schedule due to virtual learning in 2020 in response to COVID-19 may have exacerbated weight gain in adolescents with OWOB.

There are several overlapping mechanisms by which adolescents with OWOB may experience accelerated weight gain during both the summer and during COVID-19 virtual learning. Adolescents participate in more screen time and other sedentary behaviors, engage in less physical activity, and make poorer food choices when not in school [4]. The COVID-19 pandemic has also been associated with these behaviors [5]. Additionally, social distancing recommendations prevented many adolescents from participating in summer camps, sports, or other activities that typically increase their physical activity during the summer. Each of these factors is associated with poorer metabolic health and strongly predicts a greater risk for obesity [6,7], which may be detrimental to the long-term health of adolescents.

Preliminary research regarding the effects of the COVID-19 pandemic on adolescent weight gain compared body mass index (BMI) before and during the pandemic and found adolescents had a higher BMI during the pandemic [8,9,10,11,12]. However, BMI does not discriminate between total body fat percentage (TBF%) and body mass, and TBF% is a better indicator of overall health [13]. Adolescents with elevated TBF% are at greater risk for cancer, cardiometabolic diseases, and insulin resistance (IR) [14]. Additionally, IR puts adolescents at increased risk for weight gain, a poorer metabolic profile, type 2 diabetes, and other cardiometabolic diseases [15].

Additional research utilizing measures of TBF% and IR in adolescents with OWOB is needed to further assess the impact of COVID-19 on adolescent health. The present study aims to assess changes in TBF% and IR in adolescents with OWOB before and during COVID-19 to better understand the health risks related to virtual learning and social distancing during the COVID-19 pandemic. We hypothesized that adolescents with OWOB would have elevations in TBF% and IR during the COVID-19 pandemic as compared to before the pandemic and to normatively expected changes for adolescents with OWOB.

## 2. Materials and Methods

### 2.1. Design Overview

The data for this study are part of the secondary aim of a larger randomized control trial that began prior to the onset of COVID-19. This study is registered on ClinicalTrials.gov under registration number NCT02451436. Both the larger study and the COVID-19 sub-study were approved by the Institutional Review Board at the University of Alabama at Birmingham. All participants (guardians and adolescents) provided informed consent and assent for the larger study and additional consent/assent prior to sub-study data collection.

The goal of the larger study was to investigate the impact of improved sleep on adolescent insulin sensitivity and body composition. Participants were excluded from the larger study if they were actively enrolled in a weight loss intervention, had a sleep disorder or mental illness, had previously undergone bariatric surgery, or used medications that may affect sleep or weight. After enrollment, participants were randomized into either a 4-week sleep intervention or a 4-week study skills intervention, which served as an active control. Both interventions consisted of weekly, one-hour meetings focused on lifestyle modifications. Specifically, the sleep intervention concerned sleep need and the effects of media use on sleep, whereas the study skills intervention concerned effective study habits. Baseline fasting blood draws used to calculate Homeostatic Model Assessment for Insulin Resistance (HOMA-IR) and Dual Energy X-ray Absorptiometry (DXA) scans were completed during the same visit (pre-COVID1 HOMA-IR and TBF%). Follow-up blood draws were completed at ~5-weeks post-baseline (pre-COVID2 HOMA-IR), while follow-up DXA scans were completed ~6-months post-baseline (pre-COVID2 TBF%). Each of the pre-COVID measures was completed prior to March 2020. During COVID, measures were taken an average of 15 months post-baseline (Figure 1).

Following the onset of the COVID-19 pandemic, participants’ schools implemented virtual learning. Participants who were currently enrolled in virtual learning between August and November 2020 and completed a baseline appointment as part of the larger study prior to March 2020 were asked to return for an additional blood draw and DXA scan to assess TBF% and HOMA-IR in late 2020 (October–November; during-COVID HOMA-IR and TBF%) so that researchers could evaluate the effects of COVID-19-related virtual learning on adolescent health. Additional eligibility included a BMI% at or above the 85th% and access to a personal smart phone. Four participants chose to complete the during-COVID blood draw without completing the DXA. Overall, 78.6% of the present sample was part of the sleep intervention group, and 21.4% was part of the study skills group.

Within subjects, the change in HOMA-IR during the pandemic was compared to the change before COVID-19. The change in HOMA-IR during COVID-19 was also compared to demographically matched normative change data for the same developmental time period for adolescents with OWOB. To do this, normative HOMA-IR values for each participant at each analysis time point (pre-COVID1, pre-COVID-2, and during COVID) were identified using age and BMI% matching in the National Health and Nutrition Examination Survey (NHANES) population-based dataset published prior to COVID-19 [16]. These values were then used to calculate normative change and compared with COVID-19 data. Since fewer participants completed two TBF% assessments before COVID-19 compared with HOMA-IR assessments, within-subject changes in TBF% before and during COVID-19 were each compared separately to demographically matched normative changes to preserve power (Table 1). The normative change in TBF% was calculated by identifying normative values for each participant at each analysis time point (pre-COVID1, pre-COVID-2, and during COVID) matched on the participant’s age, sex, body mass index percentile (BMI%), and race using the NHANES population-based data published prior to COVID-19 [16]. Changes in total trunk fat, arm fat, and leg fat percentages were calculated but not assessed due to a lack of normatively expected changes for comparison.

### 2.2. Power Analysis

A power analysis was conducted in G*Power (v3.1). A conservative within-groups effect size of d = 0.90 was determined based on previous literature that assessed changes in adolescent percent body fat over the summer (d = 1.53) [17], BMI during COVID-19 (d = 3.48) [8], and changes in adolescent BMI over the summer (d = 0.99; 70% CI: 0.30–1.6) [18]. When using an alpha level of 0.05, a total sample size of N = 10 was indicated to achieve 80% power and allow sufficient power to detect within-subject differences in TBF% over time.

### 2.3. Measurements

Demographics. For each participant, caregivers were asked to complete a self-reported demographics questionnaire assessing date of birth, sex, race, and household income.

Total Body Fat Percent. Participants’ TBF% was assessed using dual-energy X-ray absorptiometry (DXA) with a Lunar iDXA instrument (GE Healthcare, Madison, WI, USA) with the enCORE 2011 v15 (sp2) software package. DXAs were completed at two timepoints before COVID, about 6 months apart (pre-COVID1 and pre-COVID2), and at the during-COVID visit. To assess the change in TBF% from pre-COVID1 to pre-COVID2, pre-COVID2 TBF% was subtracted from pre-COVID1 TBF% (pre-COVID1 TBF%—pre-COVID2 TBF%, Table 1). To assess the change in TBF% from pre-COVID1 to during-COVID, during-COVID TBF% was subtracted from pre-COVID1 TBF% (pre-COVID1 TBF%—during-COVID TBF%, Table 1).

Normative TBF%. Normative TBF% values were defined as expected TBF% based on both demographics and developmental timeframes and calculated using scores from a national database collected pre-COVID-19. Scores were matched to each participant based on age (in 6-month intervals), sex, BMI%, and race and calculated for pre-COVID1, pre-COVID2, and during-COVID assessments using the NHANES population-based data [16]. The normative change in TBF% from pre-COVID1 to during-COVID was calculated by subtracting the normative during-COVID TBF% from the normative pre-COVID1 TBF% (normative pre-COVID1 TBF%—normative during-COVID TBF%, Table 1). The normative change in TBF% from pre-COVID1 to pre-COVID2 was calculated by subtracting the normative pre-COVID2 TBF% from the normative pre-COVID1 TBF% (normative pre-COVID1 TBF%—normative pre-COVID2 TBF%, Table 1).

Homeostatic Model Assessment for Insulin Resistance (HOMA-IR). All participants fasted for 12 h before completing a morning blood draw that assessed glucose and insulin levels. HOMA-IR was determined using the following equation: FastGlucose(mg/dL)xFastInsulin(µU/mL)/405. To evaluate the change in HOMA-IR before COVID-19, pre-COVID2 HOMA-IR was subtracted from pre-COVID1 HOMA-IR (pre-COVID1 HOMA-IR—pre-COVID2 HOMA-IR, Table 1). To assess change in HOMA-IR from pre-COVID2 to during-COVID, during-COVID HOMA-IR was subtracted from pre-COVID2 HOMA-IR (pre-COVID2 HOMA-IR—during-COVID HOMA-IR, Table 1).

Normative HOMA-IR. Normative HOMA-IR values, defined as expected HOMA-IR values based on participant demographics and developmental time frame, were calculated using normative scores matched on each participant’s age (in 1-year intervals) and BMI% at the time of the pre-COVID2 and during-COVID blood draws. These values were identified using the NHANES population-based data published prior to COVID-19 [19]. The normative change in HOMA-IR from pre-COVID2 to during-COVID was calculated by subtracting normative values for during-COVID HOMA-IR from pre-COVID2 HOMA-IR (normative pre-COVID2 HOMA-IR—normative during-COVID HOMA-IR, Table 1).

Total Trunk, Arm, and Leg Fat Percent. Participants’ trunk fat, arm fat, and leg fat were assessed using dual-energy X-ray absorptiometry (DXA) with a Lunar iDXA instrument (GE Healthcare, Madison, WI) with the enCORE 2011 v15 (sp2) software package. DXAs were completed at two timepoints before COVID, about 6 months apart (pre-COVID1 and pre-COVID2), and at the during-COVID visit.

### 2.4. Data Analyses

All variables were assessed for normality prior to the final analyses. Sphericity was assessed prior to all analyses using repeated measures ANCOVA. Overall, all assumptions of the statistical tests were met. Paired samples *t*-tests compared (1) change in TBF% from pre-COVID1 to pre-COVID2 with normative change scores for TBF% during this same timeframe and (2) change in TBF% from pre-COVID1 to during-COVID with normative change scores for TBF% during this same timeframe (Table 1). Descriptive statistics were used to calculate the mean pre-COVID1, pre-COVID2, and during-COVID total trunk, arm, and leg fat percentages.

A repeated measures ANCOVA controlling for BMI z-score and the difference in duration between assessments compared the change in HOMA-IR from pre-COVID1 to pre-COVID2 with the change in HOMA-IR from pre-COVID2 to during-COVID. Second, repeated measures ANCOVAs compared changes in HOMA-IR from pre-COVID2 to during-COVID HOMA-IR with normative changes in HOMA-IR during this developmental timeframe. Since normative HOMA-IR data was only matched on age and BMI%, the ANCOVA included the normative variable and controlled for sex and race due to their association with HOMA-IR [19,20]. Of note, HOMA-IR analyses comparing pre-COVID and during-COVID utilized pre-COVID2 data as compared with pre-COVID1 data due to a greater number of participants with complete data for both pre-COVID2 blood draws and during-COVID blood draws as compared with those who had complete data for pre-COVID1 and during-COVID.

All analyses were conducted using IBM SPSS Statistics, version 28.0 [21]. Significance was assumed at *p* < 0.05.

## 3. Results

Participants included fourteen adolescents (M_age_ = 16.5, 57% male, 71% Black, 29% White; Table 2) with OWOB. Each participant was enrolled with a caregiver. Ten of the fourteen participants had two fasting blood draws prior to COVID-19, and five of the fourteen completed two DXAs prior to COVID-19. See Figure 2 for the participant enrollment diagram. The average time between the pre-COVID1 and the during-COVID assessments was approximately 2 years (M = 2.22, SD = 0.98).

The change in TBF% was greater from pre-COVID1 to during-COVID (M = 6.45%, SD = 4.3) as compared with the expected normative change (M = 0.54%, SD = 2.3; t(9) = 4.88, *p* = 0.001). However, there was no difference in change in TBF% for the two pre-COVID-19 assessments (M = 0.16%, SD = 1.6) as compared with expected normative change (M = −0.18%, SD = 0.63; t(4) = 0.5, *p* = 0.67; Figure 3A).

The increase in HOMA-IR between pre-COVID2 and during-COVID (M = 3.37, SD = 2.9) was greater than the increase in HOMA-IR for the two pre-COVID assessments (M = 0.78, SD = 2.9; F(1,7) = 7.89, *p* = 0.03). The increase in HOMA-IR was also greater between pre-COVID2 and during-COVID (M = 3.37, SD = 2.9) compared to normative change (M = −0.03, SD = 0.8; F(1, 7) = 12.98, *p* = 0.01; Figure 3B).

## 4. Discussion

This study demonstrates the significant impacts of the COVID-19 pandemic on the health of adolescents with OWOB, with TBF% and HOMA-IR increasing significantly during COVID-19. The change in TBF% before COVID-19 was similar to the expected normative change in adolescents with OWOB, but during COVID-19, TBF% increased by approximately six percentage points more than the normative change. By comparison, in a previous study, adolescents with OWOB were found to have increases in TBF of about 1.3% during the summer [18]. Further, adolescents’ HOMA-IR increased by ~2.5 units more during COVID-19 compared to before COVID-19 and by ~3.5 units more than the expected normative change. Given that a HOMA-IR score of 3.16 units signals significant IR for adolescents, an increase of 2.5–3.5 units during COVID-19 represents a clinically significant change [22]. Despite the small sample size (N = 14), which does limit generalizability, the study was adequately powered to detect an effect, and this clinically significant change suggests that virtual learning and social distancing during COVID-19 resulted in an accelerated increase in TBF% and HOMA-IR for adolescents with OWOB. The findings are supported by previous literature highlighting increased BMI in adolescents during COVID-19 [8,9,10,11,12]. Previous studies demonstrating associations between poorer food choices, increased screen time, sedentary behavior, decreased physical activity, and sleep disturbances in adolescents during COVID-19 provide insight into possible contributory factors related to our findings [1,5,23].

The noted Increases in TBF% and HOMA-IR are concerning for many reasons. While these findings investigate health outcomes for adolescents with OWOB and may not generalize to adolescents with healthy weight, they highlight the vulnerability of adolescents with OWOB during COVID-19. Further, when interpreting the present results, it is important to consider the context of the larger study, which worked to improve sleep and subsequently help adolescents achieve healthier HOMA-IR levels and body composition, something that should have served as a protective factor for the adolescents in the present study. Due to this, it is possible that the effects of COVID-19 on TBF% and HOMA-IR may have been even stronger for adolescents who had not previously received this intervention. Increased IR may result in hyperglycemia, hypertension, visceral adiposity, and elevated inflammatory markers, placing adolescents at increased risk for weight gain, type 2 diabetes, and other negative cardiovascular and cardiometabolic outcomes [15,24,25]. In addition, a significant acceleration in TBF% combined with an elevation of HOMA-IR may result in negative cardiovascular outcomes much earlier in life. Combined increases in adiposity and IR also predict continued obesity into adulthood [24,25], and weight gained during a brief period of time can be harder for individuals with OWOB to lose [26,27]. Therefore, adolescents with OWOB who experience exacerbated gains in TBF% and IR during COVID-19 may be at unique risk for accumulating and maintaining fat gain into adulthood and experiencing a range of negative cardiovascular and cardiometabolic health outcomes, resulting in a higher risk of premature death [28].

These results support a critical need for immediate interventions that mitigate the significant increase in TBF% and HOMA-IR during COVID-19 among adolescents with OWOB. Additionally, as the COVID-19 pandemic has not yet ended and given the possibility of new variants or novel viruses in the future, these data support interventions geared toward making the school environment safe from the spread of infection. Further, if virtual school is necessary in the future, interventions are needed to ensure children remain active, have access to healthy food, and limit screen time more effectively to mitigate negative health outcomes.

This study is the first to assess changes in TBF% and HOMA-IR before and after the pandemic, and these factors are better indicators of overall metabolic health than BMI alone. Further, these were evaluated against expected normative changes. Another strength includes a sample evenly split on sex and a majority Black. Limitations include a small overall sample. The comparison of change in TBF% before COVID-19 with expected normative change during this same time period was underpowered (N = 5) when using a conservative within-groups effect size. The study’s total sample size of N = 10 was small, which could also be a limitation. However, based on the power analysis, the sample size of 10 achieved adequate power to detect differences in the rate of change for TBF% and HOMA-IR from before to during COVID-19 as compared with expected normative change. Further, the study was strengthened by a within-subject design and the comparison of novel COVID-19 data with demographically matched normative data. Limitations also include the fact that 78.6% of adolescents in the present study had received an intervention to improve sleep duration prior to the onset of the COVID-19 pandemic, which could have indirect benefits on IR and body composition. However, previous research has demonstrated a mitigating effect of the COVID-19 pandemic on previous health behaviors taught in weight loss interventions [29], and it should be noted that the presence of a treatment effect would only serve to mute the severity of the present findings, which are notably quite strong. Finally, the effects of puberty on IR were not accounted for, and research has demonstrated that IR is influenced by pubertal development. However, the study did control the time between IR measurements, which may have helped account for pubertal changes.

Overall, this study demonstrates the significant negative health outcomes in adolescents with OWOB associated with virtual learning and social distancing due to COVID-19. Both TBF% and IR increased markedly faster during COVID-19 compared to normative data, and IR increased at a greater rate compared to before COVID-19. Exacerbations in TBF% and IR have the potential to produce enduring negative effects on adolescents’ health throughout life. Interventions to mitigate the adverse outcomes associated with COVID-19 will be critical.

## Figures and Tables

**Figure 1 children-10-01398-f001:**
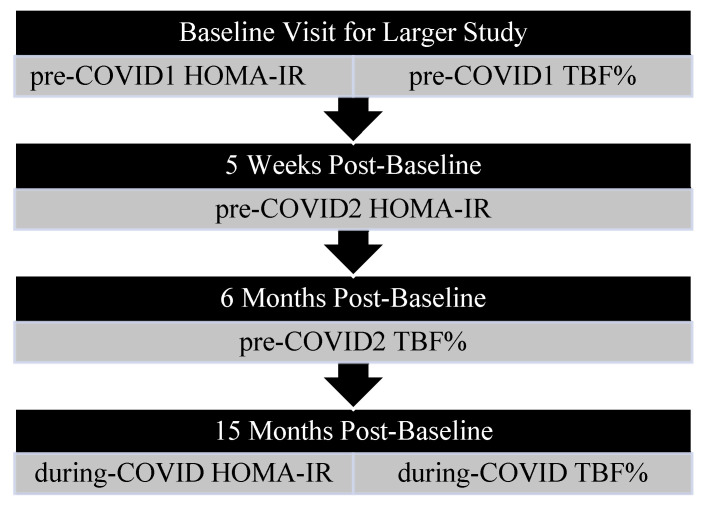
Study timeline.

**Figure 2 children-10-01398-f002:**
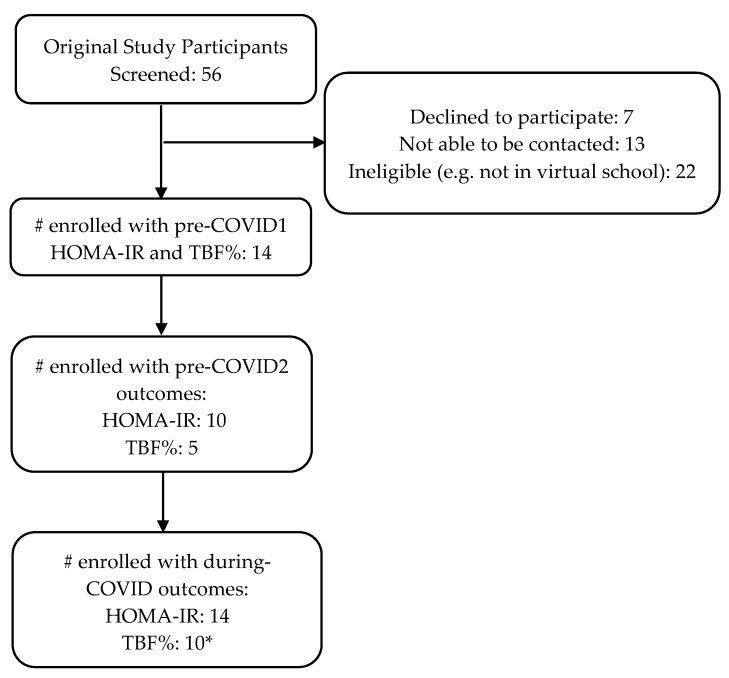
Participant enrollment diagram. * Some participants elected to complete only the blood draw for their during-COVID study visit, as TBF% was measured at a secondary location.

**Figure 3 children-10-01398-f003:**
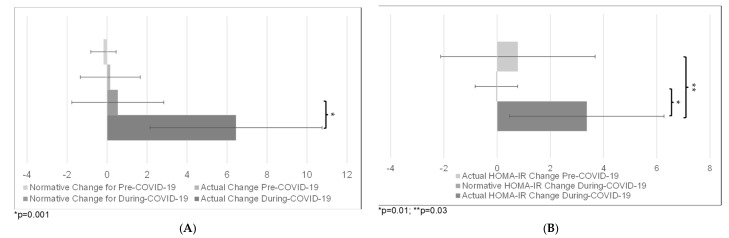
Changes in (**A**) TBF% and (**B**) HOMA-IR before/during COVID-19 compared with normative data.

**Table 1 children-10-01398-t001:** Outline of study analyses.

Analysis	Type of Test	N	Comparison Group 1	Comparison Group 2	Covariates
Change in TBF% during COVID-19 compared with normative change	Paired Samples *t*-Test	10	(pre-COVID1 TBF%—during-COVID TBF%)	(norm pre-COVID1 TBF%—norm during-COVID TBF%)	None *
Change in TBF% before COVID-19 as compared with normative change	Paired Samples *t*-Test	5	(pre-COVID1 TBF%—pre-COVID2 TBF%)	(norm pre-COVID1 TBF%—norm pre-COVID2 TBF%)	None *
Change in HOMA-IR before COVID-19 as compared with during COVID-19	ANCOVA	10	(pre-COVID1 HOMA-IR—pre-COVID2 HOMA-IR)	(pre-COVID2 HOMA-IR—during-COVID HOMA-IR)	BMI Z-Score Time between pre-2 and during-COVID visits Sex Race
Change in HOMA-IR during COVID-19 as compared with normative change	ANCOVA	10	(pre-COVID2 HOMA-IR—during-COVID HOMA-IR)	(norm pre-COVID2 HOMA-IR—norm during-COVID HOMA-IR)	Time between pre-2 and during-COVID visits Sex Race

* Normative values for TBF% and HOMA-IR were calculated using values matched to participants based on age, sex, BMI%, and race for TBF% and age and BMI% for HOMA-IR.

**Table 2 children-10-01398-t002:** Participant demographics.

Characteristics		%
Sex	Female	43.0
	Male	57.0
Household Income	<$40,000	7.1
	$40,000–59,000	28.6
	$60,000–99,999	21.4
	≥$100,000	42.9
Race	Black	71.0
	White	29.0
		**Mean (SD)**
Age (yrs)		16.5 (0.91)
BMI%	Pre-COVID1	96.9 (0.03)
BMI z-score	Pre-COVID1	2.10 (0.48)
	Pre-COVID2	2.00 (0.49)
	During-COVID	2.51 (0.34)
HOMA-IR	Pre-COVID1	3.99 (2.5)
	Pre-COVID2	4.6 (4.3)
	During-COVID	7.2 (5.6)
TBF%	Pre-COVID1	35.3 (9.8)
	Pre-COVID2	34.3 (12.2)
	During-COVID	42.3 (12.1)
Trunk Fat%	Pre-COVID1	46.94 (5.52)
	Pre-COVID2	47.27 (4.96)
	During-COVID	50.30 (4.32)
Arm Fat%	Pre-COVID1	9.82 (1.39)
	Pre-COVID2	9.63 (1.15)
	During-COVID	14.03 (10.27)
Leg Fat%	Pre-COVID1	39.85 (4.97)
	Pre-COVID2	39.36 (2.91)
	During-COVID	36.88 (3.74)

## Data Availability

The data presented in this study are available on request from the corresponding author. The data is not publicly available due to restrictions.

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
