# Peer review of "The Effects of COVID-19 Virtual Learning on Body Fat and Insulin Resistance in Adolescents with Overweight or Obesity"

_children, 2023, doi:10.3390/children10081398_

Round 1
Reviewer 1 Report
This study prospectively compared total body fat percent (TBF%) and insulin resistance (IR) among adolescents with overweight or obesity between pre-covid and during covid in the US. The authors found both TBF% and IR increased during covid. The study provided objective evidence of covid impact on adolescent obesity, by using more convincing indicators rather than BMI or body weight alone. I have just a few suggestions below.
1. Abbreviations need to be spelled out when it first appeared in text. For example, OWOB in abstract.
2. Suggest integrating line 120-130 or participants section into Design Overview to make an overview of the study. Some information also repeated in Design Overview, e.g.: line 78 to line 80.
3. What dose the sleep intervention and the study skills intervention in the study? A lifestyle modification? Because the interventions may potentially influence the study outcome, suggest describing them in method.
Reviewer 2 Report
Please complete your research with relevant data and add the requests.
The Effects of COVID-19 Virtual Learning on Body Fat and Insulin Resistance in Adolescents with Overweight or Obesity
Correct line 27 abbreviation.
Your study group is reduced by number of adolescents.
I cannot find the aim of your study.
The topic is of interest, but if you present us with a limited batch and we have no co-participation term, I'm afraid that your study is not eligible as statistical criteria. I would love to see you include more patients from the large studies you refer to and show us what conclusions emerge from your research. As it is now, it is not relevant.
Best regards.
Ask for minor changes.
